# Full-Distribution Generalization Bounds for Imitation Learning Policies

**Joseph A. Vincent**[1]    **Haruki Nishimura**[2]    **Masha Itkina**[2]    **Mac Schwager**[1]
[1]Stanford University    [2]Toyota Research Institute
{josephav, schwager}@stanford.edu
{haruki.nishimura, masha.itkina}@tri.global

**Abstract:** We present tools to bound the generalization performance of stochastic imitation learning policies deployed in novel environments. As is common in settings of robot learning from demonstrations, we assume no access to an observation or state transition model. We only have access to a small number of experimental rollouts of the policy and a performance score with which to measure the policy's success on those rollouts. Given this finite sample of performance scores, we propose a worst-case bound on the full probability distribution over the performance score. We give bounds for two kinds of performance metrics: binary task success and continuous-valued total reward. Our bounds hold with a user-specified confidence level, a user-specified tightness, and are constructed from as few rollouts in the new environment as possible. To accomplish this we build upon classical methods for constructing confidence sets without access to an underlying probability model. By defining a partial order over cumulative distributions of the performance score, we obtain confidence bounds on the *full cumulative distribution* of performance (from which one can obtain expected value bounds, quantile bounds, and numerous other bounds). We apply our approach to assess the generalization of a diffusion policy for visuomotor manipulation, where we find (potentially counter-intuitively) that the policy gives strong performance under a visually large domain shift, but weak performance under a smaller shift.

**Keywords:** Generalization, Statistical Testing, Verification

## 1   Introduction

We are interested in assessing the generalization capability of a stochastic imitation learning policy; to bound its performance in an environment different from the one in which it was trained. Our general approach is to execute a number of policy rollouts in the new environment, record the performance of each rollout, and return a confidence set for the distribution of performance in the new environment. If rollouts are independent and the environment is not time-varying, then each rollout results in an i.i.d. sample from an unknown distribution of performance. Then we use and extend classical statistical methods to place probabilistic worst-case bounds on the entire distribution of performance. Given this approach, we arrive at two research questions, (i) how to formalize the notion of a worst-case bound on a distribution, and (ii) how to obtain bounds with user specified confidence level and tightness while using as few policy rollouts as possible.

We can address the first research question immediately. Consider the cumulative distribution function (CDF) of a random variable that represents a performance metric. Given that higher performance is strictly preferable to lower performance, this implies that a CDF which is everywhere less than another CDF is strictly preferable (this is the typical notion of stochastic ordering for random variables). Thus, a partial ordering over CDFs arises where lower CDFs are preferable to higher CDFs. From this partial ordering, an upper confidence bound then serves as a worst-case distribution which is consistent with the observed data and the desired confidence and tightness.

7th Conference on Robot Learning (CoRL 2023), Atlanta, USA.

We consider two performance metrics: task success and total reward. For the task success metric the distribution of performance is Bernoulli, but with unknown success probability. In this case, an upper bound on the CDF of a Bernoulli distribution is equivalent to a lower bound on the probability of success. The total reward metric measures the reward accumulated over the course of the trajectory. In this case the distribution of performance is unknown and may be continuous, discrete, or mixed.

We make no assumptions on the definitions of user-defined task success or total reward. For instance, reward can even be time-varying within a trajectory so long as it is applied consistently across each trajectory. We also make no assumptions on the policy apart from it being stochastic. Furthermore, we make no assumptions on the underlying observation and transition models of the environment aside from them being stationary in time and do not assume access to such models. These considerations are made so that we can apply our methods to realistic settings such as robot learning from demonstrations where a mathematical model of the environment is not available. Lastly, in our experiments we use these methods to rigorously assess the generalization of a visuomotor diffusion policy [1] deployed in a new simulation environment.

## 2 Related Work

**Generalization Guarantees** In a seminal paper, Ben-David et al. [2] showed how the $\mathcal{H}\Delta\mathcal{H}$ divergence ([3]) can be used to produce a bound on the generalization of a model to a target domain [2]. Specifically, the error of a model on the target distribution can be bounded in terms of its source distribution error, $\mathcal{H}\Delta\mathcal{H}$ divergence, and the error of an optimal joint predictor over both the source and target domains. Unfortunately, the error of the optimal predictor is almost always intractable to compute.

Distributional robustness approaches assume that the exact nature of the distribution shift cannot be specified, and instead assumes that the target distribution lies within some ball around the the source distribution. However, the decision on what distance metric between distributions to use and what the radius of the ball should be is difficult. It is often the case that the Wasserstein distance metric is used, and it has been noted that the resulting generalization bounds are often too loose [4].

While each of these approaches provides bounds on the generalization error of a learned model, except in overly simplistic cases the bounds are either incomputable or vacuous. As such, we resort to testing the policy directly in the new environment.

**Binomial Confidence Sets** Confidence sets for the probability of success parameter for a binomial distribution have a long history. Confidence intervals based on normal approximations, such as the Wald (e.g. see [5]) or Wilson [6] interval, are most common in practice. However, such confidence intervals are not guaranteed to achieve the desired level of coverage. Less commonly used are confidence intervals which achieve the desired level of coverage. These methods include those of Clopper and Pearson [7], Sterne [8], Blyth and Still [9], Anscombe [10], Eudey [11], and Stevens [12]. Only the methods of [10, 11, 12] achieve *exactly* the desired coverage level. In this paper we employ the exact coverage lower confidence bound which is most clearly described in Lehmann and Romano [13]. It is worth remarking that the use of methods achieving exact coverage is nearly absent from the literature, which is echoed by the fact that none of these methods are implemented in confidence interval software packages in R, Python, Julia, or Matlab.

**CDF Confidence Sets** When determining confidence sets for the CDF of a random variable the most common method to apply is the Dvoretzky–Kiefer–Wolfowitz (DKW) inequality [14] with tight constant given by Massart [15]. However, the DKW inequality is only tight asymptotically as the number of samples approaches infinity. Smaller volume confidence sets can be achieved by methods which invert the exact Kolmogorov-Smirnov test for finite samples [16].

**Conformal Prediction** Conformal prediction is typically used to place bounds on a quantile of an unknown distribution [17] (rather than on *all quantiles simultaneously* as we do with the CDF bounds). When applying conformal prediction to the Bernoulli case, one can obtain the Clopper-Pearson [7] bound on success probability (as described in the Appendix of [18]).

# 3 Problem Formulation - Binary Success Metric

We are interested in finding confidence sets which optimally trade off between coverage, tightness, and number of samples. In this section we restrict our attention to the case of a binary performance metric. First, we introduce a lower confidence bound which optimally trades-off between coverage and number of samples. All proofs are deferred to the Appendix.

**Theorem 3.1** (Success Probability Bound). *Let $T = U + \sum_{i=1}^{n} X_i$ where $U \sim \mathcal{U}[0,1]$ and $X_i$ are i.i.d. samples from a Bernoulli distribution with unknown success probability $p$. Then*

$$\mathbb{P}(\underline{p} \leq p) = 1 - \alpha \tag{1a}$$

$$\underline{p} := \sup\{p \in (0,1) \mid F_p(t) > 1 - \alpha\}, \tag{1b}$$

*where $F_p(t)$ is the CDF of $T$, is monotonically decreasing in $p$, and is defined in the Appendix.*

Next, to control the tightness of the bound we use the notion of *shortage* (i.e., excess length),

$$\text{shortage} = \max\{p - \underline{p}, 0\}. \tag{2}$$

Since $\underline{p}$ is random, it is useful to consider expected shortage,

$$\text{ES}(p) = \mathbb{E}_p[\text{shortage}] = \int_0^p F_p(t^*(p_0))dp_0, \tag{3}$$

where $t^*$ is defined in the Appendix. This expression follows from the Ghosh-Pratt identity [19]. Expected shortage on its own is not the most useful measure of tightness because it depends on $p$, which is unknown. Instead, we assess tightness via *maximum expected shortage* (MES)

$$\text{MES} = \sup_{p \in [0,1]} \mathbb{E}_p[\text{shortage}]. \tag{4}$$

We can find MES via global optimization. Specifically, note that (3) is a mixed-monotonic function [20]. Thus, we can efficiently solve (4) using a branch-and-bound algorithm. Although the notion of MES has appeared in the literature before [21], we are the first to solve for it.

Therefore, for the case of the binary success metric we can compute confidence bounds which optimally trade-off between coverage $(1 - \alpha)$, tightness (MES), and number of samples $(n)$. Thus, given a coverage and tightness specification, the user can compute the minimum number of policy rollouts needed to obtain a lower confidence bound which meets their specifications.

# 4 Problem Formulation - Continuous Total Reward Metric

In this section we describe how to obtain an upper confidence bound on the CDF of a random variable. The CDF upper bound is constructed in such a way that given user-specified coverage rate and tightness, the minimum number of samples is used to meet these specifications.

**Theorem 4.1** (CDF Bound). *Given i.i.d. samples $X_{1:n}$ from an unknown distribution with CDF $F(x)$,*

$$\mathbb{P}(F(x) \leq \overline{F}(x) \ \forall x) \geq 1 - \alpha \tag{5a}$$

$$\overline{F}(x) = F_n(x) + \epsilon^* \tag{5b}$$

*where $F_n(x)$ is the empirical CDF and the offset term $\epsilon^*$ is defined in the Appendix. Furthermore, (5a) holds with equality if $F(x)$ is continuous.*

Note that this upper bound is strictly tighter than the bound from the standard DKW inequality.

# 5 Experiments

To demonstrate an effective use case of our framework, we deployed a trained diffusion policy [1] in an environment slightly different than it was trained on. Specifically, the environment is the *nut*

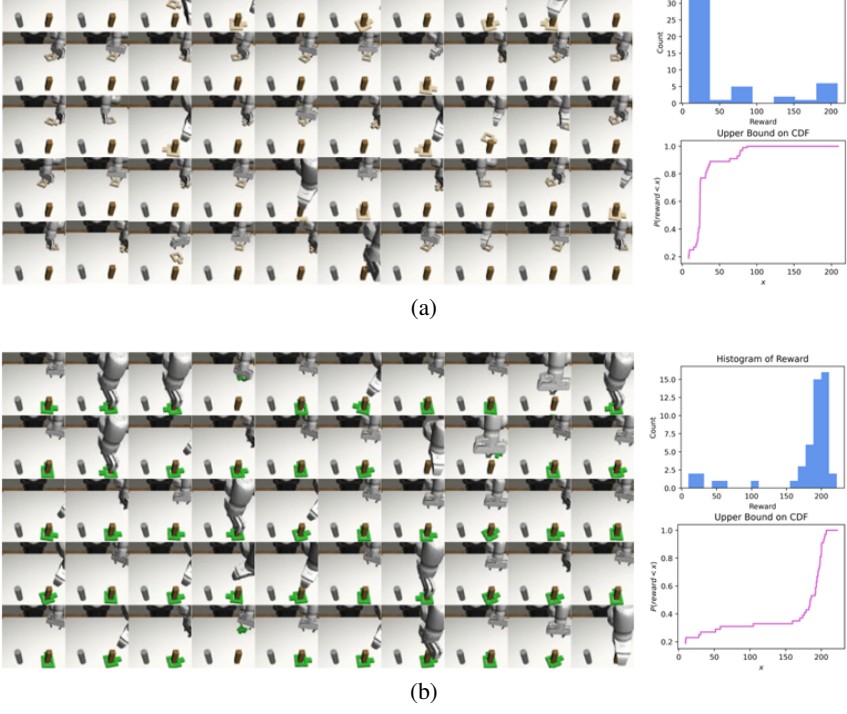

Figure 1: This figure shows the results of policy rollouts for the tan square (a) and the green square (b). Notice that, unintuitively, performance is significantly worse for the tan square than for the green square. Each subfigure includes (i) the final frame of $n = 50$ rollouts, (ii) the histogram of total reward for the rollouts, and (iii) the resulting CDF upper bound. For this experiment $\epsilon^* = 0.170$.

Table 1: Binary Performance Metric Results

| Experiment | $1 - \alpha$ | MES | $n$ | # Successes | $\underline{p}$ |
|---|---|---|---|---|---|
| tan | 0.95 | 0.118 | 50 | 9 | 0.104 |
| green | 0.95 | 0.118 | 50 | 44 | 0.798 |

*assembly* environment in the robosuite simulator [22] where the robot is tasked with placing a square nut on a wooden peg using only vision as an input. The only modification made was changing the color of the square nut from the default dark brown to (i) tan or (ii) green. We use the default binary and continuous performance metrics from the simulator.

The results in Figure 1 and Table 1 show that a practitioner can achieve relatively tight and high confidence estimates of the worst-case distribution of performance using a modest number of policy rollouts. Although the tan square seems like a smaller domain shift, the results indicate that the policy does not generalize well to this setting and generalizes better to the green square setting. This suggests that the generalization capabilities of learned policies can be unpredictable, and require a rigorous statistical approach, as we propose.

## 6 Conclusion

We developed a framework for assessing the generalization of stochastic imitation learning policies deployed in novel environments. Key to our approach is the insight that a partial ordering over performance distributions can be defined and statistical testing techniques can be used to bound the worst-case distribution of performance. We extend classical statistical techniques, deriving bounds with optimal trade-offs between coverage, tightness, and number of samples. Lastly, we are the first to give a tractable method to compute MES, which may find broad utility outside of robotics as well.

**Acknowledgments**

The authors thank the reviewers for their helpful comments.

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

# 7 Appendix

## 7.1 Notation

We let $\mathrm{bin}(k; n, p)$ and $\mathrm{Bin}(k; n, p)$ denote the Binomial probability mass function (PMF) and cumulative distribution function (CDF), respectively, with $n$ trials, success probability $p$, evaluated for $k$ observed successes. We let $\mathbb{1}(x)$ denote the indicator function which evaluates to one if $x$ is true and zero otherwise. We let $\lfloor x \rfloor$ denote the floor function which rounds the argument $x$ down to the nearest integer.

## 7.2 Binary Performance Metric

### 7.2.1 Proof of Theorem 3.1

*Proof.* This theorem follows from Corrollary 3.5.1 and Example 3.5.2 in [13]. The CDF of $T$ is defined as

$$F_p(t) = \mathrm{Bin}(\lfloor t \rfloor - 1, n, p) + (t - \lfloor t \rfloor) \cdot \mathrm{bin}(\lfloor t \rfloor, n, p). \tag{6}$$

$\square$

### 7.2.2 Monotonicity of (6)

Here we find the first derivative of the CDF in (6) with respect to $p$. First, we can write (6) as

$$F_p(t_u) = \sum_{i=0}^{\lfloor t_u \rfloor} \Delta_i \binom{n}{i} p^i (1-p)^{n-i} \tag{7a}$$

$$\Delta_i = \begin{cases} 1 & i < \lfloor t_u \rfloor \\ t_u - \lfloor t_u \rfloor & i = \lfloor t_u \rfloor \end{cases}. \tag{7b}$$

Then we can take the partial derivative with respect to $p$,

$$\frac{\partial F_p(t)}{\partial p} = \sum_{i=0}^{\lfloor t \rfloor} \Delta_i \binom{n}{i} i p^{i-1}(1-p)^{n-i}$$

$$-\Delta_i \binom{n}{i}(n-i)p^i(1-p)^{n-i-1} \tag{8a}$$

$$= \sum_{i=0}^{\lfloor t \rfloor} \Delta_i \binom{n}{i} i p^{i-1}(1-p)^{n-i}$$

$$-\Delta_i \binom{n}{i+1}(i+1)p^{(i+1)-1}(1-p)^{n-(i+1)} \tag{8b}$$

$$= (t - \lfloor t \rfloor - 1)\binom{n}{\lfloor t \rfloor}\lfloor t \rfloor p^{\lfloor t \rfloor -1}(1-p)^{n-\lfloor t \rfloor}$$

$$-(t-\lfloor t \rfloor)\binom{n}{\lfloor t \rfloor + 1}(\lfloor t \rfloor +1)p^{\lfloor t \rfloor}(1-p)^{n-\lfloor t \rfloor -1}. \tag{8c}$$

By inspection of the resulting expression one can see that $\frac{\partial F_p(t)}{\partial p} < 0$ for $p \in (0,1)$. Note that in general we do not have monotonicity over the closed interval $p \in [0,1]$.

### 7.2.3 Expected Shortage

Here we give further explanation for computing expected shortage. A useful identity we apply is

$$\mathbb{P}[\underline{p} \le p_0] = \mathbb{P}[t \le t^*(p_0)] \tag{9}$$

where $t^*(p_0)$ is the unique value that satisfies

$$F_{p_0}(t^*) = 1 - \alpha. \tag{10}$$

Then, we can compute expected shortage as

$$\mathrm{ES}(p) = \mathbb{E}_p[\text{shortage}] \tag{11a}$$

$$= \int_0^p \mathbb{P}_p[\underline{p} \le p_0]dp_0 \tag{11b}$$

$$= \int_0^p \mathbb{P}[t \le t^*(p_0)]dp_0 \tag{11c}$$

$$= \int_0^p F_p(t^*(p_0))dp_0. \tag{11d}$$

## 7.3 Continuous Reward Metric

**Definition 7.0.1** (Empirical CDF). *Consider an i.i.d. sample, $X_{1:n}$ from an unknown distribution with CDF $F(x)$. The empirical CDF (eCDF) is defined as*

$$F_n(x) = \frac{1}{n}\sum_{i=1}^{n} \mathbb{1}(X_i \le x). \tag{12}$$

### 7.3.1 Proof of Theorem 4.1

*Proof.* Consider the statistic

$$D_n^- = \sup_x \; F(x) - F_n(x). \tag{13}$$

This statistic is a one-sided Kolmogorov-Smirnov (KS) statistic. If $F(x)$ is continuous, then the distribution of $D_n^-$ does not depend on $F(x)$, i.e. it is a distribution-free statistic with the following

distribution [16],

$$\mathbb{P}(D_n^- \le \epsilon) = 1 - \epsilon \sum_{k=0}^{\lfloor n(1-\epsilon) \rfloor} w_k \tag{14a}$$

$$w_k = \binom{n}{k}(1 - \epsilon - \frac{k}{n})^{n-k}(\epsilon + \frac{k}{n})^{k-1}. \tag{14b}$$

If $F(x)$ is not continuous, then the above probability holds with $\ge$ rather than equality. Next, note that the following statements are equivalent,

$$\sup_x F(x) - F_n(x) \le \epsilon \iff F(x) \le F_n(x) + \epsilon \, \forall x. \tag{15}$$

Then we can use this result to obtain the desired result for an upper confidence bound over the entire true CDF $F(x)$. More formally, given i.i.d. samples $X_{1:n}$ from an unknown distribution with CDF $F(x)$,

$$\mathbb{P}(F(x) \le \overline{F}(x) \, \forall x) \ge 1 - \alpha \tag{16a}$$
$$\overline{F}(x) = F_n(x) + \epsilon^*, \tag{16b}$$

where $\epsilon^*$ is the solution to the equation

$$\mathbb{P}(D_n^+ \le \epsilon^*) = 1 - \alpha. \tag{17}$$

If $F(x)$ is continuous then 16a holds with equality. Given $\alpha$ and $\epsilon^*$, one can solve for the minimum number of samples needed to obtain exact coverage in the continuous case. $\square$

