# OpenReview forum: "Full-Distribution Generalization Bounds for Imitation Learning Policies"
_robot-learning.org/CoRL/2023/Workshop/OOD — OOD Workshop @ CoRL 2023_

### Official Review · Reviewer_ZAVX · 2023-10-16
**Clear and concise contribution for non-parametric conformity bounds**

**Rating:** 7
**Confidence:** 4

**Review:**

The authors propose a method that finds confidence bounds on performance for both binary and continuous tasks when deploying an algorithm on a test scenario with a fixed but different distribution than the training data. The work contains results that are consistent with the theory.

The work is very relevant to OOD generally, and while not specifically targeting OOD settings that are unique to robotics, it does target OOD on a robotics application so is relatively relevant for robotics.

The writing is clear, concise and complete.

The approach to control the tightness of the bound using the expected shortage is not previously known to the reviewer. It is a neat trick and the results seem promising.

The problem is significant, but it fails to be seen completely by not providing a comparison against other non-parametric baselines that achieve coverage (e.g. conformal prediction). An analysis of why this method is required for the optimal trade-off on samples and tightness compared to baselines would have significantly strengthened this work. Additionally, the work would benefit from a deeper analysis on the perceived performance improvement on a domain that seems further OOD. For example, training on both scenarios jointly would highlight whether the perceived human order of distribution shift holds true.

Overall, neat contribution that lacks some comparisons

---

### Official Review · Reviewer_3WZx · 2023-10-16
**Non-Parametric Method Motivating "More-Informative" Cost Functions**

**Rating:** 7
**Confidence:** 4

**Review:**

This paper presents a method for providing probability-calibrated bounds on costs and associated statistical quantities. In the case of more-informative cost functions, it extends standard scalar (e.g., mean) bounds to full-distribution specifications. The implicit testing procedure situates the results well within the OOD scope; the analysis of full cost distribution features in a nonparametric fashion builds on standard scalar-based (e.g., mean-bounding) methods. This in turn motivates the interesting idea of seeking 'more-informative' cost functions within the generalization bounds literature.

Detailed comments / thoughts:
- The paper is well-written and clearly presents the 'scaffolding' for the framework. I think that some natural comparisons would be methods like conformal prediction (e.g., [here](https://jmlr.csail.mit.edu/papers/volume9/shafer08a/shafer08a.pdf)) and PAC-Bayes results (e.g., [here](https://proceedings.mlr.press/v164/farid22a.html)). These bounds each provide valid frequentist interpretations; the former does so for more broadly general statistics and score functions, while the latter does so with uniformity over posteriors of the training data, which allows for data-dependent policy synthesis.
- The actual rates are not stated in Section 3. I am assuming that it must be $\mathcal{O}(1/\sqrt{N})$, but I would be separately curious how much tighter the exact specification for $\epsilon^*(N)$ is as compared with, e.g., Bernstein's Inequality on the empirical mean for the binary case.
- I am also curious, per the PAC-Bayes comment above, whether the bounds given generalize directly to settings beyond imitation learning. There is specific note given to the assumption of stochastic policies; however, if we instead were to assume a distribution over 'contexts' (e.g., the collection $C_i = ${initial state, obstacle configurations, dynamics}), from which we draw several instances for expert demonstration, would the bounds be invalid over the implicit domain of collections $\mathcal{C}$ (it seems like they would still be valid)?
- It seems that a one-sided KS test implicitly seeks a task-relevant aspect; namely, only detecting OOD settings that meaningfully harm performance. However, could it in principle be made double-sided by the same justification as given in this paper?

---

### Decision · Program_Chairs · 2023-10-17

**Decision:**

Accept

**Comment:**

We agree with the reviewers’ assessment that this work is technically sound and will contribute to productive, topical discussions at the 2023 Workshop on OOD Generalization in Robotics. In particular, we appreciate that your work provides both theoretical results and corroborating experiments on a problem setup squarely in the domain of OOD generalization. We recommend the authors incorporate the reviewers’ feedback into their camera-ready submission to further improve their manuscript.